# Mantle deformation records fossil convergent upwelling at Perm Anomaly

Jonathan Wolf [1,2,3] ✉, Mingming Li [4] & Barbara Romanowicz[1]

How low-velocity anomalies in the lower mantle influence convective flow, and their broader role in mantle dynamics, remain a topic of ongoing debate. One such anomaly, located roughly beneath the Russian city of Perm, is exceptionally well sampled by core-traversing seismic phases. We investigate seismic anisotropy, propagation- and polarization direction-dependent seismic wave speeds caused by deformation, within and around the Perm Anomaly at a sharp lateral resolution. Here we show a quasi-symmetric pattern of strong seismic anisotropy delineating the boundary of the Perm Anomaly, and linear streaks of strong anisotropy pointing towards the boundary. Geodynamic modeling experiments suggest such patterns of anisotropy are signatures of convergent upwelling mantle flow; 'frozen-in' fossilized deformation of ancient origin is a plausible explanation supported by geodynamic modeling. Furthermore, we detect seismic anisotropy within the anomaly indicative of internal deformation, though it is substantially weaker than that observed near the edges.

Earth's deepest mantle hosts two large, equatorial, and antipodal regions with seismic velocities a few percent lower than the depth-averaged value, known as Large Low-Velocity Provinces (LLVPs)[1–3]. Smaller low-velocity features also exist[4,5] consistent across tomographic models, the largest of which was identified by Lekic et al.[4] and dubbed Perm Anomaly. What governs the genesis and evolution of these anomalies remains debated. Previous research suggests a link between the Perm Anomaly and large igneous provinces[4,6–8], and proposes that its origin is tied to past mantle flow[7,8]. However, there is a lack of observational evidence constraining flow at the Perm Anomaly. Bridging this gap is essential for understanding deep-mantle dynamics and the potential origin of surface volcanism.

Flow-induced deformation, in particular in D″, the lowermost 200–300 km of the mantle, can manifest itself through seismic anisotropy, a property whereby shear-wave velocities depend on the polarization and/or propagation direction of the wave[9–11]. Measurements of seismic anisotropy thus provide snapshots of Earth's interior deformation and mantle flow field. Due to the dense broadband seismic networks in Europe, which provide excellent coverage of core-refracted waves[12] around the Perm Anomaly, this anomaly is uniquely well suited to investigate the nature of such low-velocity features and their role in mantle convection, a topic that remains actively debated.

Strong seismic anisotropy is frequently observed in the deep mantle near low-velocity features[13–19], including the Perm Anomaly[20,21], and commonly interpreted as evidence for a transition from horizontal to vertical flow. However, while the strength of anisotropy can often be quantified, the inference of flow directions is typically indirect and relies on multiple assumptions[11], for example, on the dominant deformation mechanism or the dominant slip systems of D″ minerals, about which there is currently no consensus[9–11]. As a result, beyond the detection of strong anisotropy at D″ edges of LLVPs, mantle flow adjacent to these high-deformation zones has remained poorly resolved.

In this study, we present evidence for the presence of regularly organized, horizontally elongated deformation streaks with strong seismic anisotropy in the vicinity of the Perm Anomaly and strong seismic anisotropy delineating its boundary. Through geodynamic modeling at global and regional scales, we propose that these anisotropy features are due to convergent flow toward the Perm Anomaly, implying an upwelling flow component.

[1]Department of Earth and Planetary Science, University of California, Berkeley, CA, USA. [2]Miller Institute for Basic Research in Science, Berkeley, CA, USA. [3]Department of Earth and Planetary Sciences, University of California, Santa Cruz, CA, USA. [4]School of Earth and Space Exploration, Arizona State University, Tempe, AZ, USA. ✉e-mail: wolf@ucsc.edu

## Results

### Deep mantle deformation mapped by core-refracted waves

Shear-wave measurements, commonly used to investigate seismic anisotropy[22,23], rely on the fact that shear waves that travel through anisotropy divide into two orthogonally polarized components that travel at different speeds. The splitting intensity (*SI*, see Methods) quantifies the magnitude of shear-wave splitting that a seismic phase undergoes. To detect D″ anisotropy, splitting intensities of SKS and SKKS, or PKS and SKKS phases (Fig. 1a, c) are commonly compared[13,16,24–26]. This method is effective because the lateral raypath separation distance between SKS and SKKS (or PKS and SKKS) is large (between 600 and 850 km; Fig. S1) in the lowermost mantle, while their raypaths are nearly identical in the upper mantle, and the bulk of the lower mantle is nearly isotropic[24,27] (Fig. 1a).

The region around the Perm Anomaly is densely sampled by core-refracted SKS, SKKS, and PKS waves (Fig. 1b and d). We sort |*δSI*| values in 30° directional swaths (e.g., Fig. 1d). For every 2° × 2° geographic bin, we consider the maximum absolute average *δSI* value across all directional swaths for which coverage is obtained, as minimum estimate of the anisotropy strength within that bin and call this value *Max*(|*δSI*|). As single-station differential splitting results can show some spread for any particular region (e.g., due to noise)[28], any azimuthal swath for which fewer than five measurements are obtained is discarded. The results are robust to alternative averaging schemes and associated uncertainties (Supplementary Figs. S2 and S3).

If *KS pairs sampled D″ at the same angle to the vertical direction and were influenced by upper mantle anisotropy in an identical way, differential SKS (or PKS)-SKKS splitting would indicate a lateral gradient in lowermost mantle anisotropy. However, the different angles at which SKS (or PKS) and SKKS travel through D″ can lead to splitting differences for a uniformly anisotropic layer[25,29]. Therefore, differential splitting is influenced by both the magnitude and gradient of D″ anisotropy. Following previous studies that conducted detailed synthetic tests[25,29,30], we only interpret measurements with |*δSI*| ≥ 0.4 as being indicative of lowermost mantle anisotropy, accounting for the fact that the slight raypath differences of *KS pairs in the upper mantle can lead to minor differences in splitting[30].

### Regions with strong deformation

We observe seismic anisotropy within the Perm Anomaly, shown as anisotropic feature A in Fig. 2 (inset). While differential splitting in the center of the anomaly is generally weak, it is the strongest along the

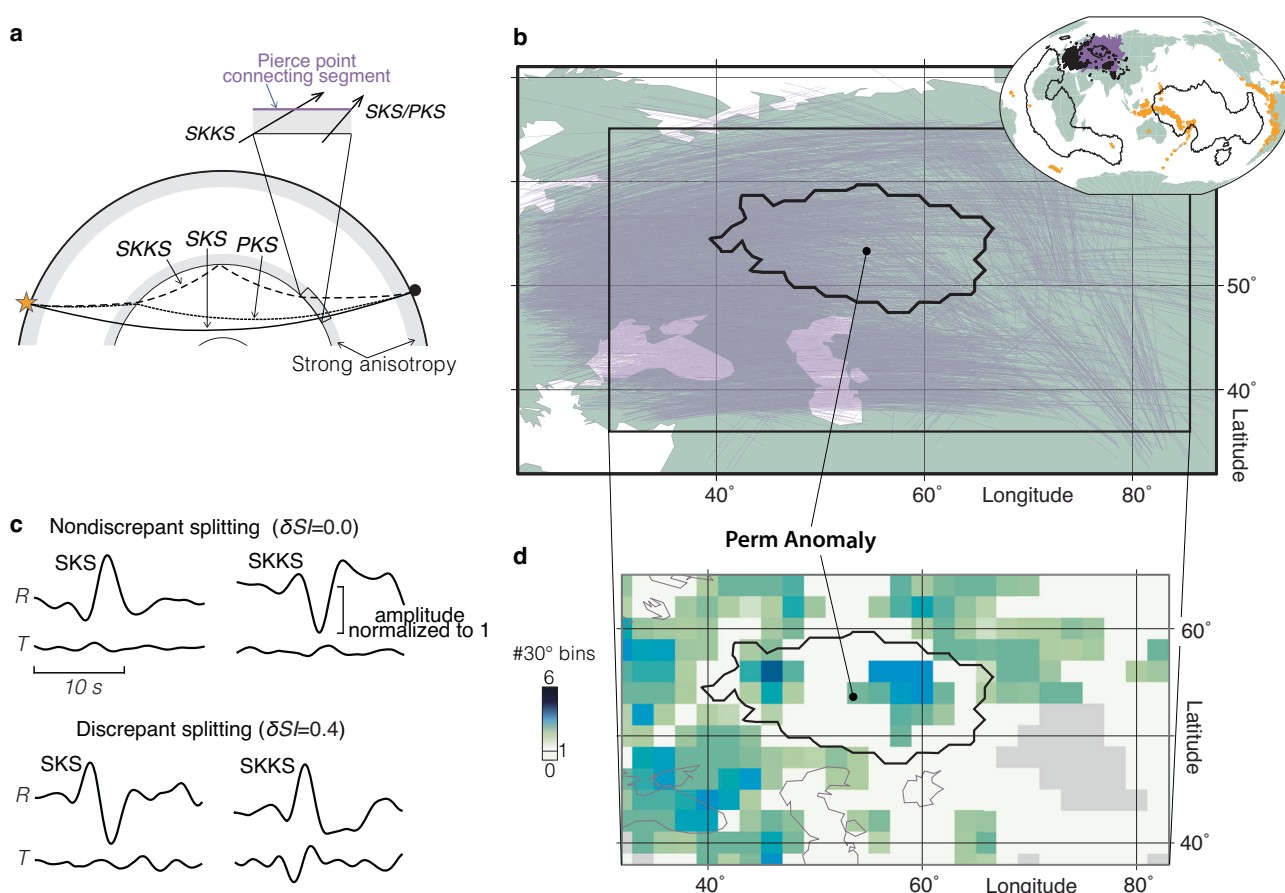

**Fig. 1 | Ray coverage in our study region. a** Earth cross-section showing SKS (solid line), SKKS (dashed line), and PKS (dotted line) raypaths. Significant seismic anisotropy can be found in the upper and lowermost mantle (light gray). Zoom-in: Schematic explanation of pierce point connecting segments (violet). Pierce point connecting segments link the SKKS pierce point through the core-mantle boundary with the SKS (or PKS) pierce point through the upper D″ boundary, assumed here to be 250 km above the core-mantle boundary. Black arrows through D″ indicate SKKS and SKS raypaths. **b** Pierce point connecting segments for all source-receiver combinations viewed from above (violet lines). The Perm anomaly from cluster analysis[4]. Black lines show where three out of five models indicate low velocities in a cluster analysis[4] (for models that contribute to this analysis, see Supplementary Information). Inset: Events (yellow stars) and stations (black circles) used in this study. **c** Top: SKS (left) and SKKS (right) radial (R) and transverse (T) velocity seismograms for a nondiscrepant splitting pair with differential splitting intensity *δSI* = 0.0. The traces are from an event that occurred on July 21, 2007, and was recorded at station OBN. Bottom: Same as top panel for a discrepant SKS-SKKS pair (*δSI* = 0.4) and traces from an event that occurred on April 14, 2008, recorded at station OBN. **d** Number of 30° directional intervals with more than five measurements sampled per 2° × 2° geographic bin (legend). Gray indicates no ray coverage. Figure panels were made using Obspy[48] and GMT[49].

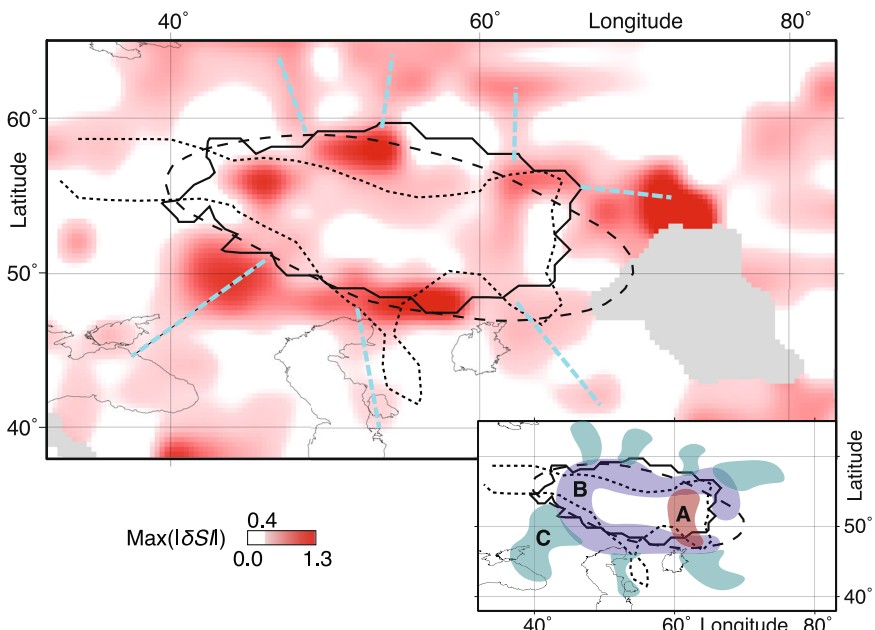

**Fig. 2 | Splitting results.** Maximum splitting intensity values, $Max(|\delta SI|)$ (legend), for all directional bins. Gray indicates no ray coverage. Black lines indicate locations of the Perm Anomaly determined by a cluster analysis (solid line where majority of models show low velocities)[4], tomography model GLAD-M25[50] (dashed line for S velocity outline at $7.23\frac{km}{s}$ at 2800 km), and by an S-ScS residual travel time analysis[51] (dotted line at 0.7% travel time anomaly). Linear anisotropic features are indicated by dashed blue lines. Inset: Naming convention of anisotropic features, A (red), B (blue), C (turquoise). For a version of this figure without smoothing and for corresponding bootstrap uncertainties, see Supplementary Fig. S3. Figure was created using GMT[49].

edge of the Perm Anomaly (Fig. 2 inset, anisotropic feature B), but it is not possible to determine whether it is straddling the edge of the Perm anomaly or on one side of it, given that the location of the anomaly's boundary is somewhat dependent on the tomographic study (Fig. 2). In many locations along the Perm Anomaly edge, the lowermost mantle is sampled from only a single 30° directional interval (Fig. 1d), but the observed $Max(|\delta SI|)$ values are higher than those in the directionally better-sampled regions farther from the edge (Fig. 2), indicating that the anisotropy is strongest along this edge.

We observe elongated anisotropic features (anisotropic features C, Fig. 2 inset) that are directed toward the edge of the Perm Anomaly, from here on called deformation streaks. Along these streaks south of the Perm Anomaly, the splitting intensity signature is consistent across bins when sampled from similar directions (Supplementary Figs. S4 and S5), indicating coherent deformation along these structures. The potential deformation streaks to the north are less well sampled (Figs. S4 and S5), making it more difficult to draw firm conclusions and to rule out the possibility that some of these features may partly reflect limited ray coverage.

Lateral variations in strain (and deformation) or the presence of several mineral phases lead to lateral differences in shear-wave splitting. The dominant constituent in the lower mantle is bridgmanite (Br), which undergoes a phase transition to post-perovskite (pPv) near the CMB, although the depth of this phase transition depends on temperature and chemical composition[31,32]. Seismic velocities around the Perm Anomaly are relatively fast, implying below-average temperatures and a relatively shallow Br-pPv transition, while the opposite is likely within the anomaly. Therefore, differences in the anisotropic signature inside and outside of the Perm anomaly may reflect changes in the depth of the phase transition, but the observed variations within these two regions are most likely explained by deformation.

### Implications for convective flow near the Perm Anomaly

Strong seismic anisotropy along the western edge of the Perm Anomaly was reported previously[20]. Our observations show that this observation extends to nearly its entire circumference (anisotropic feature B) and

suggest a ring-like pattern of deformation along the edges of the Perm Anomaly. We also observe streaks of stronger seismic anisotropy pointing toward the edge of the Perm Anomaly (anisotropic feature C). Such features are only seismically detectable here due to the exceptional ray coverage in our study region. Our detection of anisotropic feature A, located in the interior of the Perm Anomaly (Fig. 2), also adds to the limited number of documented cases of seismic anisotropy within low-velocity structures of the lowermost mantle[16,33].

Narrow elongated zones of elevated strain aligned with present-day mantle flow as well as concentrated strain near the edges of low-velocity structures, potentially associated with a component of upwelling flow, have been predicted by previous geodynamic modeling[34–37]. In such models, the Perm Anomaly is generally flowing towards the northwest at the present-day (Supplementary Fig. S6). Time-dependent geodynamic models[7] with plate motion history imposed suggest that flow around the Perm Anomaly evolved from convergent during its early formation to a predominantly westward advection since ~ 150 Ma.

To elucidate the relationship between the mantle flow field and observed deformation, we employ regional geodynamic models in a spherical box under varying flow conditions. A convergent upwelling regime provides the simplest physical explanation for the symmetry and orientation of the observed streaks. This hypothesis is tested in our "Type-1" models (Methods), for which we impose a divergent flow field at the surface; this boundary condition drives downwelling along the domain walls and induces convergent flow toward the center at depth (Supplementary Fig. S7). We initialize the model with a global, 25–50 km thick layer of intrinsically dense material atop the CMB. The convergent basal flow sweeps this material into a central thermochemical structure. While flow away from this structure is predominantly lateral, significant upwelling occurs near its edges (Fig. 3a). Tracking finite strain accumulation via tracers[35], we find that this setup reproduces all primary anisotropic features (A, B, C) observed in the data (Fig. 3a and Supplementary Fig. S8). Notably, while the vertical extent of the high-strain streaks depends on the initial dense layer thickness, a parameter unconstrained by our seismic data, their lateral

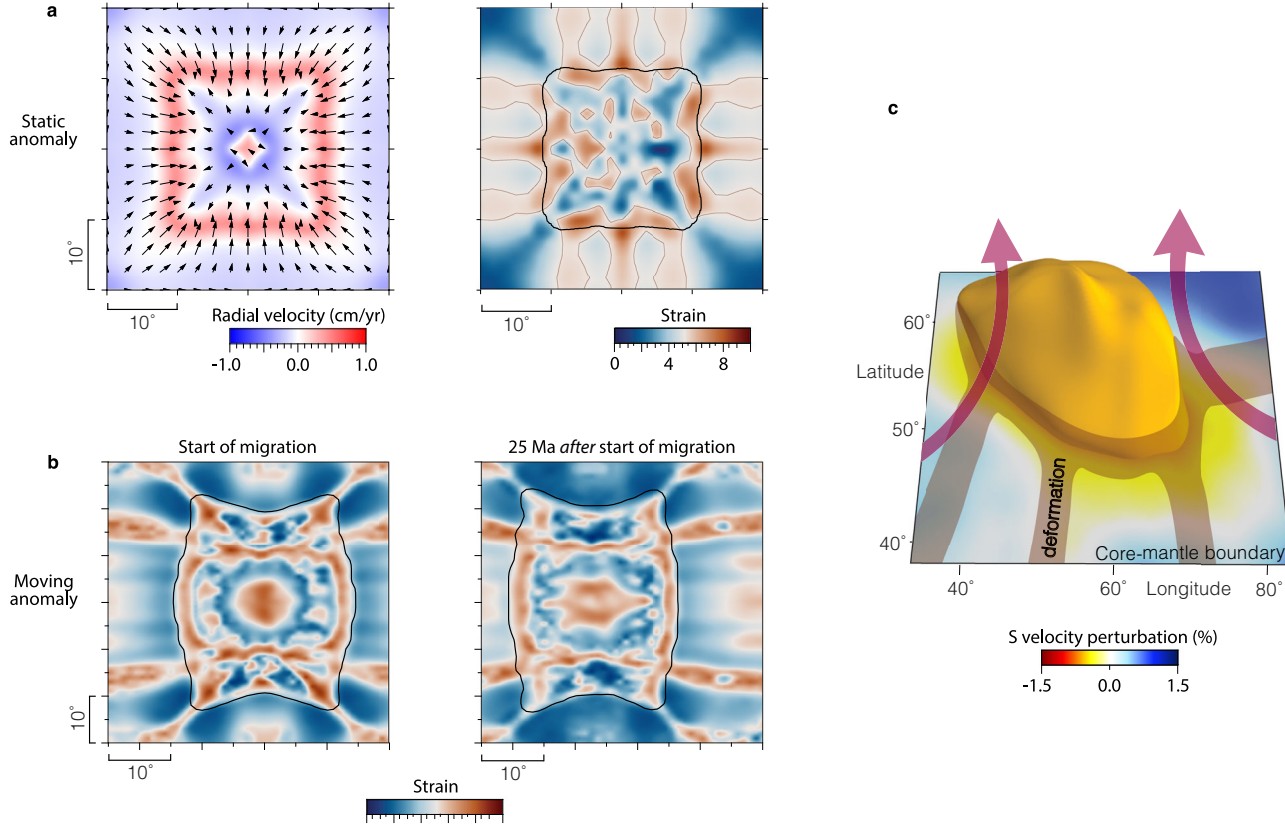

**Fig. 3 | Geodynamic results and interpretation. a** Left: Radial (red-to-blue colors, legend) and lateral (black arrows) flow velocity 45 km above the core-mantle boundary obtained from a Type-1 geodynamic model (Online Methods). Right: Geodynamically derived depth-averaged strain (legend, outline at 5.2) in the lowermost 112.5 km of the mantle. Black lines show the extent of the thermochemical anomaly. **b** Depth-averaged strain in the lowermost 120.0 km of the mantle (legend) for a Type-2 geodynamic model that includes a westward motion of the Perm Anomaly. Left: Onset of westward migration. Right: After 25 Myr of westward motion. Only results at the central parts of the model domain are shown in (**b**), which are less sensitive to boundary conditions. Black lines show the extent of the thermochemical anomaly. **c** The Perm Anomaly shown as a −0.8% isocontour from seismic tomography[50] (legend). Interpreted flow directions are shown as violet arrows, and zones of strong deformation in brown shading. Figure was created using GMT[49].

spatial pattern remains robust. Furthermore, the model also predicts strain accumulation within the thermochemical structure itself, consistent with the internal anisotropy (Feature A) detected within the Perm Anomaly (Fig. 2).

For the Type-2 regional models (Methods), we initially impose divergent flow at the top surface to generate convergent flow in the lowermost mantle, again producing streaks of strong strain. Subsequently, we switch the top boundary to free-slip, applied stress-free conditions to the east and west side boundaries, and impose a constant westward velocity at the CMB. In this model, the thermochemical structure is advected westward by the velocity imposed at the CMB. Although the anomaly itself is transported westward, the previously developed high-strain pattern persists for several tens of Myr (Fig. 3b and Supplementary Fig. S9a, b) before evolving substantially by 100 Myr (Supplementary Fig. S9c). If mantle flow were unidirectional and uniform at all depths, this strain pattern would remain unchanged for >100 Myr. Instead, gradual modification occurs because unidirectional motion is driven only at the CMB, leaving overlying regions subject to intrinsic mantle dynamics. We note that neither end-member scenario—uniform flow at all depths or strictly CMB-driven motion—fully represents actual mantle flow around the Perm Anomaly since 100 Ma. In reality, slabs sinking slabs through the mantle and the viscosity contrast between upper and lower mantle have an influence on flow and deformation.

Our geodynamic modeling results suggest that a convergent flow field around the Perm Anomaly induces (1) high strains within the anomaly, (2) intense deformation surrounding the anomaly's edges, and (3) linear streaks of high strain pointing toward those edges. These predicted patterns correspond to our seismic observations of anisotropy features A, B, and C, respectively. Specifically, the strong deformation at the edges of the Perm Anomaly marks a transition in mantle flow direction from lateral to vertical (Fig. 3). Similar transitions from horizontal to vertical flow have been proposed in earlier studies of seismic anisotropy at the margins of LLVPs[14,16,20,38]. Furthermore, the streaks of concentrated deformation directed toward the anomaly align with patterns inferred from global seismic tomography[35], while the roughly symmetric distribution of anisotropy around the edges further supports a scenario of convergent flow. We point out that the strength of the convergent flow may be overestimated due to the use of a regional modeling domain and the Boussinesq approximation. However, our main objective here is to capture the characteristic strain pattern rather than the full spectrum of Earth's complexities. A convergent flow field appears to offer the most parsimonious and natural explanation for the observed patterns and symmetry of deformation.

In contrast to a scenario of active convergence, previous global models[7,35] indicate that present-day mantle flow around the Perm Anomaly is predominantly westward. Our results show that the extent to which—and the timescale over which—the previously developed strain pattern is preserved depends on how closely the Perm Anomaly undergoes simple westward translation (Fig. 3b and Supplementary Fig. S9). Therefore, our interpretation of a convergent signature does not require the flow field to be currently convergent; the observed anisotropy could instead represent a "frozen-in" fabric preserved from past geodynamic conditions. While our seismic results alone cannot distinguish between active and fossilized flow, given that the Perm

Anomaly has likely existed for at least ~ 150 million years, it is probable that preserved anisotropy plays a critical role. A past convergent regime is dynamically favorable for the accumulation of thermochemical anomalies and may have characterized the anomaly's early formation prior to ~ 150 Ma[7].

Seismic anisotropy at the base of major plumes associated with active hotspots have been linked to present-day strain associated with the transition from dominantly horizontal to vertical flow[38–40], although lateral resolution has been poor. In contrast, the Perm Anomaly is not associated with a currently active hotspot and its root does not appear to contain a mega ultra-low velocity zone[4] as do several major active plumes. Our high-resolution study is indicates that it represents the remnant of a past or present upwelling[7,8], with our geodynamic simulations suggesting at thermochemical origin. The high resolution at which we were able to conduct our study, may possibly be extended in the future, to investigate the roots of some well-illuminated active mantle plumes, or through targeted experiments to better understand the dynamics and evolution of focused deep mantle upwellings.

## Methods

### Shear-wave splitting measurements

We use event-station pairs for earthquakes with moment magnitudes of 5.9 or greater that occurred between January 1, 2000, and October 1, 2024, for which both SKS (or PKS) and SKKS phases (Fig. 1a) sample D″ within latitudes 32° to 83° and longitudes 38° to 65°, near the Perm Anomaly. These data are part of a massive global dataset, collected from 25 datacenters, for which networks and their corresponding citations are provided in Supplementary Notes 1 and 2.

SKS, PKS, and SKKS waves are fully SV-polarized upon their P-to-S conversion at the CMB on the receiver side leg of the raypath. If seismic anisotropy is present on along the upwards raypath through the mantle, energy is split from the radial component $R(t)$ to the transverse component $T(t)$, which takes the shape of the radial component time derivative $R\prime(t)$[22,23]. Both the magnitude of splitting and the transverse energy shape are assessed using the splitting intensity ($SI$)[41], expressed as:

$$SI = -2 \frac{T(t)R\prime(t)}{|R\prime(t)|^2}. \tag{1}$$

We use differential SKS-SKKS and PKS-SKKS splitting intensity measurements from Wolf et al.[30] in this study, which were determined with SplitRacerAuto[42] from data bandpass-filtered between 6 and 25 s. Only $SI$ measurements with formal 95% confidence intervals[41] smaller than ± 0.4 are retained, following previous work that conducted detailed synthetic experiments[25,30,43]. Only absolute splitting intensities below 1.5 are used because the additivity of the splitting intensity may break down for larger values[29].

These $\delta SI$-values are binned using an equal area 2° × 2° grid. Binning is conducted separately for each 30° directional interval for which five or more measurements are obtained (if fewer measurements are obtained, the interval is discarded). As shear-wave splitting varies dependent on polarization direction of the wave, the maximum absolute $\delta SI$-value across all directional swaths is selected for each bin, which provides an estimate of the maximum differential splitting that we obtain evidence for.

### Geodynamic modeling

For the thermochemical calculation conducted in this study, we solve the following conservation equations of mass, momentum and energy under the Boussinesq approximation:

$$\nabla \cdot \mathbf{u} = 0 \tag{2}$$

$$-\nabla P + \nabla \cdot (\eta \, \dot{\boldsymbol{\epsilon}}) + Ra(T - BC)\hat{\mathbf{r}} = 0 \tag{3}$$

$$\frac{\partial T}{\partial t} + (\mathbf{u} \cdot \nabla)T = \nabla^2 T + H, \tag{4}$$

where $\mathbf{u}$ is velocity, $P$ is dynamic pressure, $\eta$ is viscosity, $\dot{\boldsymbol{\epsilon}}$ is the strain-rate tensor, $T$ is temperature, $\hat{\mathbf{r}}$ is the unit vector in the radial direction, $B$ is the buoyancy number, $C$ is composition, $t$ is time, and $H$ is internal heating rate. These conservation equations are solved using the CitcomCU code[44].

The Rayleigh number $Ra$ is defined as:

$$Ra = \frac{\rho_0 \, g \, \alpha \, \Delta T \, R^3}{\eta_0 \, \kappa}, \tag{5}$$

where $\rho_0$, $g$, $\alpha$, $\eta_0$, and $\kappa$ are the reference density, reference gravity acceleration, reference thermal expansivity, reference viscosity, and reference thermal diffusivity, respectively. $R$ is the radius of the Earth, and $\Delta T$ is the reference temperature, which equals to the temperature increase from the surface to the CMB. The values used in this study for these reference parameters are: $R = 6371$ km, $\rho_0 = 3300$ kg/m³, $\alpha = 1 \times 10^{-5}$ K⁻¹, $\kappa = 1 \times 10^{-6}$ m²/s, $g = 9.8$ m/s², $\Delta T = 3000$ K, and $\eta_0 = 5 \times 10^{21}$ Pa s. With these parameters, $Ra = 5 \times 10^7$.

The model is both internally and basally heated with an internal heating rate of $H = 100$. Temperature is fixed at $T = 0$ on the top surface and $T = 1$ on the CMB, with insulating side boundaries.

The viscosity law expressed by $\eta = \eta_r \exp[A(0.6 - T)]$, where $A = 6.91$ is the non-dimensional activation energy. $\eta_r$ is the viscosity pre-factor controls the depth dependence of viscosity. In this study, $\eta_r = 0.3$, 0.03, and 1.0 at depth ranges of 0–100 km, 100–670 km, and 670 km–CMB, respectively.

We perform two types of models with different model size and boundary conditions. For the first type of model, or Type-1 models, we impose a divergent velocity field at the top surface, $v_\theta = -v_0 \cos(\theta) \sin(\phi)$ and $v_\phi = -v_0 \cos(\phi)$, where $\theta$ and $\phi$ are co-latitude and longitude, respectively, and $v_0 = 2$ cm/yr. All other boundaries have free-slip velocity boundary conditions. The model domain covers the whole-mantle depth, with a non-dimensional radius ranging from 0.55 on the CMB to 1.0 on the surface, and covers a longitude and a cola-titude range of 70°–110° and 70°–110°, respectively. There are 64, 64, and 128 elements in the longitudinal, co-latitudinal, and radial directions, respectively, resulting in an average lateral resolution of ~ 38.0 km near the CMB and a radial resolution of ~ 22.5 km. A global layer of intrinsically dense material is initially placed at the bottom of the model, which has a thickness of 50 km and a buoyancy number of 0.8. Here, the buoyancy number is defined as $B = \Delta\rho_c/(\rho_0\alpha\Delta T)$, where $\Delta\rho_c/\rho_0$ is the intrinsic density anomaly. So, B = 0.8 is equivalent to an intrinsic density anomaly of 2.4% with $\alpha = 1 \times 10^{-5}$ K⁻¹ and $\Delta T = 3000$ K. For comparison, we also perform other Type-1 models in which the global layer has a thickness of 25 with a buoyancy number of 2.0 (or an intrinsic density anomaly of 6%) (Supplementary Fig. S5). The advection of the composition field is simulated using the ratio tracer method[45]. The divergent flow field at the top surface results in a convergent flow in the lowermost mantle, which pushes the initially global layer of intrinsically dense material into a thermochemical pile at the center of the model domain in the lowermost mantle.

In the second type of models, or Type-2 models, we first use the same boundary conditions as Type-1 models, e.g., with divergent flow field at the top surface. After the formation of a thermochemical pile at the model center, we make the top surface free-slip and the west and east side boundaries stress-free, and we impose a constant westward unidirectional flow field with a speed to 2 cm/yr. To minimize potential boundary effects, we expand the domain in both longitudinal and co-

latitudinal directions, each spanning a range of 30°–150°. We use 128 elements in both the longitudinal and co-latitudinal directions and preserve the same lateral resolution as Type-1 models. In the radial direction, 128 elements are employed, with the grid refined to 15 km resolution in the lowermost 150 km of the mantle. Like for Type-1 models, we introduce a global layer of intrinsically dense material at the bottom of the model as initial condition, which has a thickness of 60 km and a buoyancy number of 0.5. We advect the compositional field using the absolute tracer method[45].

In addition to active tracers that are used to simulate the compositional field, we also use passive tracers to track the strain information of materials. Passive tracers are continuously placed at 300 km depth above the CMB and also on the CMB occupied by the intrinsically dense material at each timestep. Each tracer begins with zero strain, and we accumulate its strain along its trajectory using the method described in detail in McNamara (2003)[46] and Li et al.[35]. For simplicity, we do not consider recrystallization process, which may lead to reset of strains[35]. The strains of passive tracers are projected to grid points when we plot the strain maps as in Fig. 3a, b.

## Data availability
All data used in this study are publicly available and were collected and pre-processed as part of ASU's global data collection system (http://adept.sese.asu.edu/) for their global data products project (http://swat.sese.asu.edu). More details are specified in the Supplementary Information.

## Code availability
Shear-wave splitting measurements were conducted using SplitRacerAuto[42]. For the precise version we used, we refer to Wolf et al.[47], who have made this version publicly available. Geodynamic simulations use the CitcomCU code[44], available at https://github.com/geodynamics/citcomcu.

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

## Acknowledgements

J.W. was funded by the Miller Institute for Basic Research in Science at UC Berkeley.

## Author contributions

Conceptualization: J.W.; Data analysis: J.W.; Geodynamic modeling: M.L.; Methodology: J.W., M.L.; Visualizations: J.W., M.L.; Writing initial draft: J.W.; Review and editing: J.W., M.L., B.R.; Resources: J.W., M.L., B.R.

## Competing interests

The authors declare no competing interests.
