## [Transparent Peer Review file · Nature Communications]

Mantle deformation records fossil convergent upwelling at Perm Anomaly

Corresponding Author: Dr Jonathan Wolf

Version 0:

Reviewer comments:

Reviewer #1

(Remarks to the Author)

Review of 'Mantle deformation suggests convergent upwelling flow at the Perm Anomaly' submitted by Wolf et al to Nature Communications

This well-written and well-illustrated manuscript documents 'seismic anisotropy, propagation- and polarization direction-dependent seismic wave speeds caused by deformation, within and around the Perm Anomaly at an unprecedented lateral resolution'. This reveals 'a quasi-symmetric pattern of strong deformation' around the Perm Anomaly. Regional geodynamic models with a symmetric setup in a spherical box produce a symmetric pattern of deformation, which is proposed to explain much of the seismic observations. In the geodynamic model, surface divergence results in downwelling along the sides of the domain and upwelling in the centre of the domain. Based on new seismic observations (and models) and on geodynamic models, it is proposed that the Perm Anomaly is a local center of convergent and upwelling mantle flow.

The seismic observations show that there is strong seismic anisotropy around most of the circumference of the Perm Anomaly (anisotropic feature B), to the possible exception of the eastern boundary, where there is no coverage. There is also an anisotropic anomaly within the Perm Anomaly (anisotropic feature A) and strong linear deformation generally perpendicular to the boundary (anisotropic feature C), which are best established to the south of the boundary. It is unclear how the direction of the strong linear deformation (away from or towards the boundary) could be established. The text suggests that the deformation is towards the boundary.

The tested hypothesis is that low-velocity features have a role in mantle convection. The results of global mantle flow models, presented in Supplementary Figure 3, suggest that the mantle under Perm is generally flowing towards the northwest, which is consistent with previous work that suggested a westward migration of the Perm Anomaly from ~200 million years ago (Fig. 5e of Flament et al., 2017). The models shown in Supplementary Figure 3 suggest that mantle flow could be deflected around the anomaly and could be slower through the anomaly than around it (in map view just above the core-mantle boundary). There also seems to be more strain to the north and east of the Perm Anomaly. These results suggest that the low-velocity Perm Anomaly may play a limited role in mantle convection: at present, the flow would mostly be driven by downwellings to the east and south of the Perm Anomaly (Figure S3, Figure 5e of Flament et al., 2017; McNamara, 2022), with some local modification around the Perm Anomaly.

The results of these models are discarded based on the low resolution of global seismic tomography models. Instead, regional geodynamic models in a spherical box are presented, with finite elements of comparable dimensions to the global mantle flow models of Li et al. (2024), but independent of lower-resolution tomographic models. A symmetric divergent flow field that depends on latitude and longitude is imposed at the surface. This leads to symmetric downwelling along the walls of the box, and symmetric upwelling from the centre and base of the box (Figure S4). It is unclear how time is handled. Are the simulations instantaneous, or is strain calculated over time?

Can the flow pattern predicted by the regional mantle flow models (Figure 3a and Figure S4) be reconciled with the expected global mantle flow (Figure S3)?

It is unclear whether the results of regional mantle flow models can be used to gain insight into global mantle flow. This is well established for slab dynamics (Peng and Liu, 2023). Plume flux is much greater in a box model than it is in a global

model. In the absence of adiabatic cooling, plume flux is greater using the Boussinesq approximation than using the extended Boussinesq or the truncated anelastic liquid approximation. Overall, the model setup (symmetric divergence) arguably predermines the outcome of the experiments (symmetric downwelling on the walls of the domain and upwelling from the base of the domain), overestimates the influence of plumes on flow (regional model, Boussinesq approximation), and does not represent the role that a realistic subduction scenario could have on the formation of the Perm Anomaly. Given these limitations, it is surprising that the presented geodynamic models are put on the same level as seismic observations in the abstract ('Geodynamic modeling of Perm as a thermochemical anomaly shows that such streaks typically align with mantle flow, providing a novel means of mapping flow in the deepest mantle.')

The 'downwelling flow along all side boundaries' is proposed to be 'consistent with the presence of mantle downwellings around the Perm Anomaly as suggested by previous studies (Flament et al., 2017; He et al., 2021)' (L. 89-90). This statement is surprising. Flament et al. (2017) wrote 'In this tectonic scenario, subduction to the west of the Perm-like anomaly ceases when the Mongol-Okhotsk Ocean closes 150 Myr ago.', and showed that downwelling is expected to have changed over time and that downwelling is not expected around the Perm Anomaly at present-day when it is restricted to the east and south of the Perm Anomaly (Figure 5 of Flament et al., 2017). The material presented by He et al. (2021) suggests seismically fast material in the bottom 500 km of the mantle predominantly to the east of the Perm Anomaly. Is the presented model applicable to the Perm Anomaly?

The results of the mantle flow models are proposed to be consistent with the seismic observations, and the role of radial mantle flow (which is likely overestimated by the models) is emphasised (Fig. 3c, L. 101-102, L. 108). Could the anisotropy features around the Perm Anomaly that this work maps in unprecedented detail be explained by a scenario other than mantle flow convergence? Can a scenario of westward migration of the Perm Anomaly be ruled out on the basis of these observations?

The link between the Perm Anomaly and a mantle plume is emphasised in the text (e.g. L.6-10), however, there is no evidence for a recent volcanic eruption associated with the anomaly, and no seismic evidence for a plume associated with the anomaly (L. 122-126). Can a 'frozen-in' upwelling (L. 125) be expected to have a large effect on mantle flow?

Note that contrary to what Flament et al. (2017) suggested, the Perm Anomaly could not have caused the Emeishan LIP that formed near the equator at around 260 Ma (Torsvik and Domeier, 2017).

The presented geodynamic models are generic. How do the model results compare to seismic observations around other basal mantle structures (even though the Perm Anomaly might be the best sampled anomaly for anisotropy)?

Could the new seismic anisotropy data be used to test whether the Perm Anomaly has been fixed or mobile over time? The results presented in Figure S3 (and their consistency with the results of Flament et al., 2017) suggest that the Perm Anomaly could be mobile and entrained by large-scale mantle flow.

The presented seismic anisotropy results are intriguing. Comparing these results to geodynamic models is not straightforward. One avenue would be to analyse global mantle flow models that predict quasi-symmetric pattern of strong deformation around the Perm Anomaly, which seems to be consistent with the observations introduced in this contribution (see Fig. 3 of Creasy et al., 2020).

Nicolas Flament, Wollongong, 15 October 2025

References

Creasy, N., Miyagi, L. and Long, M.D., 2020. A library of elastic tensors for lowermost mantle seismic anisotropy studies and comparison with seismic observations. *Geochemistry, Geophysics, Geosystems*, 21(4), p.e2019GC008883.

McNamara, A.K., 2022. Mobile mantle could explain volcanic hotspot locations.

Peng, D. and Liu, L., 2023. Importance of global spherical geometry for studying slab dynamics and evolution in models with data assimilation. *Earth-Science Reviews*, 241, p.104414.

Torsvik, T.H. and Domeier, M., 2017. Correspondence: Numerical modelling of the PERM anomaly and the Emeishan Large Igneous Province. *Nature communications*, 8(1), p.821.

Reviewer #2

(Remarks to the Author)

In this manuscript, the authors present anisotropy in the Perm low velocity anomaly in the low mantle, using a high density of core-refracted phases. The large number of station event pairs enable an unprecedented high lateral resolution investigation of such a feature with the described methods. The observed details of symmetric deformation patterns along the boundary of the anomaly, as well as linear features in radial fashion, allow the authors to draw conclusions on mantle flow patterns that can be reproduced with geodynamic modelling. Since the resolution of these deep features has been relatively poor and,

thus, conclusions remain a matter of debate, this work presents an important addition that should be interesting to the readers and relevant for future investigation in this and other areas.

In general, this manuscript has a good and concise structure and the ideas presented are setup, developed and interpreted in a logical way that is easy to follow. The figures are very clear and well-structured. I have a few suggestions and questions of clarification. I would like to mention that I am not an expert on the geodynamic modelling part of this study, therefore, it would be good to have another reviewer with expertise in that field.

Below I will list individual comments in the order they appear in the manuscript:

I. 35: "lateral raypath reparation distance between SKS and SKKS (or PKS and SKKS) is large"
Even though the distances are indicated by the length of the segment lines in Figure 1b, it would be helpful to briefly mention the approximate distance here.

I. 39: "30° directional swaths"
This value is probably a balance between having a tight bin to have the best resolution possible, and a required minimum number of ray paths in that orientation. Were there any tests to see if tighter (if possible) or broader bins would change the result significantly? Also, were there any tests to check how the results might vary if different limits for the 30°-bins are being chosen (e.g., 40°-70°, 50°-80° or something like that)?

I. 39: "2°x2°"
A slightly similar comment to the one above. Have there been any tests with different grid sizes?

Figure 1:

- "blue" – I do not know if this is an issue with the document, but the blue lines in Figures a and b appear purple in my version.
- "black line majority of analyzed models show low velocities" – this explanation could be made clearer in my view by using the same caption as in Supplementary Figure 2a ("Black lines show where three out of five models indicate low velocities in a cluster analysis"). If not, it would at least be good to have a colon behind "black line".
- "Bottom: Same as left panel" – I think this should have been "Bottom: Same as top panel"

Figure 2:

- Is there a reason the surface has been smoothed? Supplementary Figure 1 shows the 2°x2° grid pixels, so a similar thing could be reproduced here (just to avoid the fact that features are misinterpreted due to changes in their shapes as a result of the smoothing).
- Here and also especially in Supplementary Figure 2: are there any uncertainties on the values? It could be helpful to see anything like that to estimate the robustness of the features presented and discussed in this study.

Figure 3:

- Have there been models with different shapes of the feature, other than what appears to be a square one? Like in the comment above, this again could be interesting for the robustness estimation of the features.

I. 162: "Δt" – this should be "ΔT".

David Schlaphorst
Lisbon, 27/10/2025

Version 1:

Reviewer comments:

Reviewer #1

(Remarks to the Author)

Review of 'Mantle deformation records fossil convergent upwelling at Perm Anomaly' submitted to Nature Communications by Jonathan Wolf, Mingming Li, and Barbara Romanowicz

The authors have carefully considered reviewer comments and made changes accordingly, including running new mantle flow simulations that suggest that the pattern of anisotropy associated with the initial upwelling of the Perm Anomaly could be retained after at least 25 million years of lateral motion (Figure 3b). These new experiments are an elegant and appropriate way to address my original comments. Is there a reason why the results are only shown after 25 million years, and not for later times? The size of the new ('Type-2') models is larger than the size of the original ('Type-1') models, making it possible to consider a phase of lateral advection. The half-width of the model (~6,600 km) and the migration velocity (2 cm/yr or 22 km/Myr) determine the duration for which a phase of lateral advection can be considered (maximum 6,600/22 = 300 Myr). In practice, that duration is shorter because edge effects would be likely to affect the flow and the results after some time (depending on the flow condition on the left-hand side; an open flow boundary condition is, in principle, possible but arguably not necessary in this case). However, 25 million years is much shorter than 300 million years. It would be helpful to present and discuss the results of the models after a longer phase of lateral advection – ideally up to 150 million

years. At which stage (if any) would a pattern of anisotropy associated with lateral advection be expected to develop? How long could 'frozen-in' patterns of anisotropy associated with the upwelling of a Perm-like anomaly be retained for? Could this information be used to speculate about the evolution of the Perm Anomaly (its possible history of upwelling and lateral advection)?

In the abstract, consider changing 'likely' to 'possible'

L. 44–45: consider rephrasing to 'The results are robust to alternative averaging schemes and associated uncertainties (Supplementary Figures S2 and S3).'

L. 126: consider changing 'subtleties' to 'properties'

Nicolas Flament, Wollongong, 22 January 2026

Reviewer #2

(Remarks to the Author)

This is the revised version of the manuscript that I have reviewed before, so I will mainly focus on the authors' responses to my comments and the changes in general. I think the authors put effort into answering the suggestions of both reviewers to improve the manuscript. If those are sufficient, I can only confidently answer for my suggestions; for the other reviewer's comments it would be better to have their feedback. It is good to see that the additional tests shown in the added supplementary figures added to the robustness of the study overall and I am happy with the revised version.

David Schlaphorst
Lisbon, 24/01/2025

Version 2:

Reviewer comments:

Reviewer #1

(Remarks to the Author)

Review of 'Mantle deformation records fossil convergent upwelling at Perm Anomaly' submitted to Nature Communications by Jonathan Wolf, Mingming Li, and Barbara Romanowicz

The authors have elegantly addressed my remaining comment by showing results after 48 Myr and 98 Myr of imposed westward flow at the CMB (Fig. S9b–c). I look forward to seeing this work published. If they wish to do so, the authors could moderate the text on L. 112–122 at the proof stage. Two end-member mantle flow scenarios are mentioned: 'unidirectional motion only at the CMB' (L. 120–121; scenario implemented in the model) or 'mantle flow [] unidirectional and uniform at all depths' (L. 117–118). Some text clarifying that neither scenario is likely to be representative of mantle flow from ~100 Ma around the Perm Anomaly would be helpful. Notwithstanding the complex interaction of sinking slabs with mantle upwelling, 'mantle flow [] unidirectional and uniform at all depths' seems incompatible with the viscosity contrast between the upper mantle and lower mantle inferred from studies of the geoid and post-glacial rebound: lower mantle flow is likely to be slower than upper mantle flow.

Nicolas Flament, Wollongong, 20 February 2026

We have revised our manuscript, now called "Mantle deformation records fossil convergent upwelling at Perm Anomaly" for Nature Communications. We are grateful for the very constructive comments from the reviewers, which have helped us to significantly clarify and strengthen the manuscript. Below we have reproduced the comments from you and the reviewers verbatim in *black, italicized* text, with our responses in green color and regular font. Where useful, we denote line numbers in the revised manuscript where changes were made, which are in green font in the new manuscript's revised text.

Two main points were raised by the reviewers:

The first point concerns how well we interpret seismic observations using geodynamic models. We are grateful to Reviewer #1 (R1) for raising a critical point regarding the link between seismic observations and predictions from global geodynamic models. R1 questioned whether the observed deformation patterns could represent 'frozen-in' texture formed >100 Ma, persisting despite potential present-day westward advection indicated by independent geodynamic models (including R1's plate-reconstruction-based and our own tomography-based models shown see Supplementary Figure S6). Previously, we had attributed this discrepancy to resolution limits, hypothesizing that seismic data captured local flows unresolved by global models.

However, prompted by R1's suggestion, we performed further investigations. We found that deformation streaks generated by convergent upwelling can indeed persist for over 100 million years, even after the onset of westward advection. Consequently, the scenario of fossilized, 'frozen-in' deformation successfully reconciles our high-resolution seismic observations with the broader geodynamic constraints. We are very grateful to R1 for guiding us toward this interpretation.

The second point pertains to the uncertainty and robustness of our measurements in light of different potential averaging schemes, bin sizes, and azimuthal widths. We have explored this thoroughly, implementing each of Reviewer #2's (R2's) suggestions. The new results, now shown in Supplementary Figures S2 and S3, demonstrate that our findings are robust across a wide range of data processing strategies. In fact, we could not identify any alternative choices that would alter three main anisotropic features interpreted in Figure 2 of the main manuscript. We have also computed formal uncertainties for our anisotropy map, which are generally small compared to the magnitude of the interpreted $\delta S/I$ values (see new Supplementary Figure S3b).

We believe the manuscript has been significantly improved by incorporating the constructive feedback from the reviewers.

Reviewer #1 (R1)

Review of 'Mantle deformation suggests convergent upwelling flow at the Perm Anomaly' submitted by Wolf et al to Nature Communications

This well-written and well-illustrated manuscript documents 'seismic anisotropy, propagation- and polarization direction-dependent seismic wave speeds caused by deformation, within and around the Perm Anomaly at an unprecedented lateral resolution'. This reveals 'a quasi-symmetric pattern of strong deformation' around the Perm Anomaly. Regional geodynamic models with a symmetric setup in a spherical box produce a symmetric pattern of deformation, which is proposed to explain much of the seismic observations. In the geodynamic model, surface divergence results in downwelling along the sides of the domain and upwelling in the centre of the domain. Based on new seismic observations (and models) and on geodynamic models, it is proposed that the Perm Anomaly is a local center of convergent and upwelling mantle flow.

We thank R1 (Dr. Nicolas Flament) for the summary of the paper, general comments above and the constructive, detailed suggestions below, which greatly improved the manuscript.

The seismic observations show that there is strong seismic anisotropy around most of the circumference of the Perm Anomaly (anisotropic feature B), to the possible exception of the eastern boundary, where there is no coverage. There is also an anisotropic anomaly within the Perm Anomaly (anisotropic feature A) and strong linear deformation generally perpendicular to the boundary (anisotropic feature C), which are best established to the south of the boundary. It is unclear how the direction of the strong linear deformation (away from or towards the boundary) could be established. The text suggests that the deformation is towards the boundary.

We thank R1 for this accurate summary of our findings.

The tested hypothesis is that low-velocity features have a role in mantle convection. The results of global mantle flow models, presented in Supplementary Figure 3, suggest that the mantle under Perm is generally flowing towards the northwest, which is consistent with previous work that suggested a westward migration of the Perm Anomaly from ~200 million years ago (Fig. 5e of Flament et al., 2017). The models shown in Supplementary Figure 3 suggest that mantle flow could be deflected around the anomaly and could be slower through the anomaly than around it (in map view just above the core-mantle boundary). There also seems to be more strain to the north and east of the Perm Anomaly. These results suggest that the low-velocity Perm Anomaly may play a limited role in mantle convection: at present, the flow would mostly be driven by downwellings to the east and south of the Perm Anomaly (Figure S3, Figure 5e of Flament et al., 2017; McNamara, 2022), with some local modification around the Perm Anomaly. The results of these models are discarded based on the low resolution of global seismic tomography models. Instead, regional geodynamic models in a spherical box are presented, with finite elements of comparable dimensions to the global mantle flow models of Li et al.

(2024), but independent of lower-resolution tomographic models. A symmetric divergent flow field that depends on latitude and longitude is imposed at the surface. This leads to symmetric downwelling along the walls of the box, and symmetric upwelling from the centre and base of the box (Figure S4). It is unclear how time is handled. Are the simulations instantaneous, or is strain calculated over time?

We thank R1 for this comment. We agree that global mantle flow results suggest the Perm anomaly is primarily advected passively to the west or northwest (this is, interestingly, the case for R1's plate-reconstruction-based models as well for ours that are based on global tomography). Our initial modeling demonstrated that purely passive advection fails to reproduce the observed seismic anisotropy, which requires a history of convergent flow. Previously, we had attributed this discrepancy to resolution limits, hypothesizing that seismic data captured local flows unresolved by global models.

To further investigate this, we have performed new geodynamic simulations that explicitly model a temporal transition from convergent to unidirectional flow. We found that the strain patterns established during the initial convergent phase remain during the subsequent unidirectional motion. Thus, while our seismic observations indicate that the Perm anomaly originates from a center of convergence and upwelling, the anisotropy signal does not constrain the timing of the flow.

We therefore, after these additional simulations, have come to the conclusion that the only flow scenario that reconciles all seismic and geodynamic constraints is one that includes fossilized, 'frozen-in' deformation. This scenario, in contrast to our initial argument, is fully consistent with present day westwards or northwestwards mantle flow indicated by various geodynamic models, and therefore resolves this discrepancy. We thank R1 for pointing us in this direction and refer to lines 109-137 of the revised manuscript which describe this in detail.

The flow field in the global models is instantaneous, and the flow field in the regional models is time-dependent by solving the conservation equations of mass, momentum and energy. Tracers are introduced to the model domain to track strains. We have clarified the setup of these models in the thoroughly revised method section of the manuscript (lines 188-215).

Can the flow pattern predicted by the regional mantle flow models (Figure 3a and Figure S4) be reconciled with the expected global mantle flow (Figure S3)?

Yes, it can. As described above, fossilized, 'frozen-in' deformation from past flow can reconcile our high-resolution seismic observations with both the regional and global geodynamic models. We now describe this in detail at 129-137 of the revised manuscript, and have also changed the manuscript accordingly in multiple other places (for example, abstract, lines 87-93, 109-114, and 142-143).

It is unclear whether the results of regional mantle flow models can be used to gain insight into global mantle flow. This is well established for slab dynamics (Peng and Liu, 2023). Plume flux

is much greater in a box model than it is in a global model. In the absence of adiabatic cooling, plume flux is greater using the Boussinesq approximation than using the extended Boussinesq or the truncated anelastic liquid approximation. Overall, the model setup (symmetric divergence) arguably predermines the outcome of the experiments (symmetric downwelling on the walls of the domain and upwelling from the base of the domain), overestimates the influence of plumes on flow (regional model, Boussinesq approximation), and does not represent the role that a realistic subduction scenario could have on the formation of the Perm Anomaly. Given these limitations, it is surprising that the presented geodynamic models are put on the same level as seismic observations in the abstract ('Geodynamic modeling of Perm as a thermochemical anomaly shows that such streaks typically align with mantle flow, providing a novel means of mapping flow in the deepest mantle.')

We thank the reviewer for highlighting these model limitations and the distinctions between regional and global flow approximations. We agree that our box models, using the Boussinesq approximation, simplify the physics of plume flux and do not capture all dynamic complexities. We have added a discussion of these limitations and their implications at lines 124-128.

However, we emphasize that the primary purpose of these models is not to predict the full flow field ab initio, but to test a specific hypothesis: Does convergent upwelling flow produce the specific deformation patterns observed in our seismic data? While the flow geometry in our box model is indeed imposed (and thus 'predetermined'), the resulting development of crystallographic preferred orientation (CPO) and seismic anisotropy is not. The key finding is that this specific flow geometry reproduces the complex seismic observations with striking accuracy. While we cannot strictly rule out all other scenarios, the convergent upwelling model offers the simplest physical explanation for the symmetry and orientation of the observed streaks. We have clarified this distinction between predicting flow vs. testing deformation mechanisms in the revised text (e.g., lines 94-108).

The 'downwelling flow along all side boundaries' is proposed to be 'consistent with the presence of mantle downwellings around the Perm Anomaly as suggested by previous studies (Flament et al., 2017; He et al., 2021)' (L. 89-90). This statement is surprising. Flament et al. (2017) wrote 'In this tectonic scenario, subduction to the west of the Perm-like anomaly ceases when the Mongol-Okhotsk Ocean closes 150 Myr ago.', and showed that downwelling is expected to have changed over time and that downwelling is not expected around the Perm Anomaly at present-day when it is restricted to the east and south of the Perm Anomaly (Figure 5 of Flament et al., 2017). The material presented by He et al. (2021) suggests seismically fast material in the bottom 500 km of the mantle predominantly to the east of the Perm Anomaly. Is the presented model applicable to the Perm Anomaly?

Thanks for the comments. We have removed this wrong statement in the revised manuscript.

The results of the mantle flow models are proposed to be consistent with the seismic observations, and the role of radial mantle flow (which is likely overestimated by the models) is emphasised (Fig. 3c, L. 101-102, L. 108). Could the anisotropy features around the Perm

Anomaly that this work maps in unprecedented detail be explained by a scenario other than mantle flow convergence? Can a scenario of westward migration of the Perm Anomaly be ruled out on the basis of these observations?

We thank the reviewer for these insightful questions regarding the interpretation of our results.

Regarding the flow scenario: While we acknowledge that our modeling of the Perm Anomaly as a center of convergent mantle flow does not strictly exclude all other possibilities, and that the real Earth is undoubtedly more complex than our first-order geodynamic models, we contend that a convergent flow scenario offers the most parsimonious physical explanation for the symmetry of the deformation and orientation of the observed deformation streaks. While other mechanisms might contribute to the deformation, it is challenging to find an alternative scenario that reconciles the full suite of observed anisotropy patterns as naturally as the convergent model. In response to this comment, we have expanded our discussion (e.g., lines 94-108, 115-128) to explicitly state that while convergence is the most robust explanation for the observations, we do not rule out additional complex deformation mechanisms.

Regarding westward migration: We agree that our proposed flow scenario is not in conflict with a potential westward migration of the Perm Anomaly. In fact, as demonstrated by our new geodynamic models, treating the Perm Anomaly as a "frozen-in" feature from a fossilized upwelling provides a framework that satisfies both the seismological observations and the geodynamic constraints of present-day flow models. We have added a detailed discussion of this possibility and how it aligns with migration scenarios in the revised manuscript (e.g., lines 129-137).

The link between the Perm Anomaly and a mantle plume is emphasised in the text (e.g. L.6-10), however, there is no evidence for a recent volcanic eruption associated with the anomaly, and no seismic evidence for a plume associated with the anomaly (L. 122-126). Can a 'frozen-in' upwelling (L. 125) be expected to have a large effect on mantle flow? Note that contrary to what Flament et al. (2017) suggested, the Perm Anomaly could not have caused the Emeishan LIP that formed near the equator at around 260 Ma (Torsvik and Domeier, 2017).

We appreciate this feedback. Regarding the influence of a "frozen-in" upwelling on the broader mantle flow field, while we did not explicitly test these dynamic feedbacks in our local models, our global models (e.g., Figure S6) suggest that such effects would be limited in magnitude.

Additionally, we agree with the caution regarding the connection to Large Igneous Provinces (LIPs). In the revised manuscript, we have removed the discussion linking the Perm Anomaly to specific LIP events. We now simply note in general terms that previous studies have associated the Perm Anomaly with LIP formation, avoiding speculative correlations to keep the focus on our robust seismic results (lines 5-9).

The presented geodynamic models are generic. How do the model results compare to seismic observations around other basal mantle structures (even though the Perm Anomaly might be the best sampled anomaly for anisotropy)?

We appreciate this intriguing point. In the revised manuscript, we now note that strong seismic anisotropy is indeed observed near LLVPs, as well as at the deep mantle roots of major hotspot plumes (e.g., lines 17-23, 138-140). However, we emphasize that previous studies generally lacked the lateral resolution required to resolve fine-scale features, such as the deformation streaks observed here (lines 138-140).

Furthermore, as discussed in the revised text (lines 143-146), we propose that with sufficient ray sampling, the techniques applied in this study are applicable to other key deep mantle regions, including LLVPs, their margins, and plume roots.

Could the new seismic anisotropy data be used to test whether the Perm Anomaly has been fixed or mobile over time? The results presented in Figure S3 (and their consistency with the results of Flament et al., 2017) suggest that the Perm Anomaly could be mobile and entrained by large-scale mantle flow.

This is an interesting point. When considered independently of global geodynamic flow predictions, our seismic anisotropy data are consistent with two non-unique scenarios: a stationary Perm Anomaly currently undergoing convergent flow, or a migrating anomaly that has preserved the strain fabric of past convergence. Consequently, seismic anisotropy alone is insufficient to distinguish between a stationary or mobile evolutionary history. We now mention this at lines 134-136 of the revised manuscript.

While expanding the seismic anisotropy analysis to a larger area could theoretically confirm whether flow is primarily westward, this is practically difficult given current data constraints. Distinguishing these scenarios would require either ϕ -dt measurements associated with lowermost mantle anisotropy or the detection of deformation streaks in adjacent areas. To obtain ϕ -dt measurements, we tried ScS and Sdiff but were not successful, given the strict data requirements for them (not every source-receiver pair is suitable for ScS and Sdiff splitting measurements due to the initial source polarization; in fact, most are unsuitable). Additionally, *KS sampling is much poorer just east of the Perm Anomaly, making it impossible to resolve deformation streaks at the required lateral resolution.

The presented seismic anisotropy results are intriguing. Comparing these results to geodynamic models is not straightforward. One avenue would be to analyse global mantle flow models that predict quasi-symmetric pattern of strong deformation around the Perm Anomaly, which seems to be consistent with the observations introduced in this contribution (see Fig. 3 of Creasy et al., 2020).

We thank R1 for this helpful comment. Interestingly, Li et al. (2024) find that on a global scale, the two large-low velocity provinces (LLVPs) are under convergent flows and our previous

geodynamic models predict similar linear streaks of deformation around the LLVPs, which is broadly consistent with observations of seismic anisotropy about the LLVPs. More high-resolution seismic images of anisotropy and more refined mantle flow models in local regions would greatly improve our understanding on lowermost mantle dynamics. We have mentioned these points in the revised manuscript (lines 87-93). However, as R1 said, comparing geodynamic modeling results with seismic observations is not straightforward. More discussions about this type of comparison is presented in our previous paper (Li et al., 2024) and is also referenced in the revised manuscript.

Creasy et al.'s Figure 3 is indeed interesting. It illustrates upwelling flow along the margins of the Perm Anomaly and LLVPs, which provides important context. However, a direct comparison with our anisotropy results remains limited because their study focuses on flow velocity fields rather than accumulated finite strain, which is the primary control on the observed anisotropy. We have now incorporated this citation into the revised manuscript (line 89).

Li, M., et al. (2024). "Flow and Deformation in Earth's Deepest Mantle: Insights From Geodynamic Modeling and Comparisons With Seismic Observations." *Journal of Geophysical Research: Solid Earth* 129(12)

Nicolas Flament, Wollongong, 15 October 2025

References

*Creasy, N., Miyagi, L. and Long, M.D., 2020. A library of elastic tensors for lowermost mantle seismic anisotropy studies and comparison with seismic observations. *Geochemistry, Geophysics, Geosystems*, 21(4), p.e2019GC008883.*

McNamara, A.K., 2022. Mobile mantle could explain volcanic hotspot locations.

*Peng, D. and Liu, L., 2023. Importance of global spherical geometry for studying slab dynamics and evolution in models with data assimilation. *Earth-Science Reviews*, 241, p.104414.*

*Torsvik, T.H. and Domeier, M., 2017. Correspondence: Numerical modelling of the PERM anomaly and the Emeishan Large Igneous Province. *Nature communications*, 8(1), p.821.*

Thanks again for your comments and suggestions, which have greatly helped improve the paper. Thanks for these references as well.

Reviewer #2 (R2)

In this manuscript, the authors present anisotropy in the Perm low velocity anomaly in the low mantle, using a high density of core-refracted phases. The large number of station event pairs enable an unprecedented high lateral resolution investigation of such a feature with the described methods. The observed details of symmetric deformation patterns along the boundary of the anomaly, as well as linear features in radial fashion, allow the authors to draw conclusions on mantle flow patterns that can be reproduced with geodynamic modelling. Since the resolution of these deep features has been relatively poor and, thus, conclusions remain a matter of debate, this work presents an important addition that should be interesting to the readers and relevant for future investigation in this and other areas.

In general, this manuscript has a good and concise structure and the ideas presented are setup, developed and interpreted in a logical way that is easy to follow. The figures are very clear and well-structured. I have a few suggestions and questions of clarification. I would like to mention that I am not an expert on the geodynamic modelling part of this study, therefore, it would be good to have another reviewer with expertise in that field.

We thank R2 (Dr. David Schlaphorst) for the general comments above and the constructive, detailed suggestions below, which have helped us to significantly improve the manuscript.

Most importantly, R2 requested additional information on the uncertainty and robustness of our measurements. We have now performed all of the suggested tests and added three new Supplementary Figures (S1–S3) in response. These figures demonstrate the robustness of our results across a large range of averaging schemes, show uncertainties, and enable readers to more easily and independently assess the reliability of our measurements.

Below I will list individual comments in the order they appear in the manuscript:

I. 35: “lateral raypath separation distance between SKS and SKKS (or PKS and SKKS) is large” Even though the distances are indicated by the length of the segment lines in Figure 1b, it would be helpful to briefly mention the approximate distance here.

Good suggestion. We now mention at lines 35-36 of the revised manuscript that the lateral separation distance is always between 600 and 850 km. We also had a figure handy that showed the lateral separation distance as a function of epicentral distance for both phase pairs, which we have included as the new Supplementary Figure S1.

I. 39: “30° directional swaths”

This value is probably a balance between having a tight bin to have the best resolution possible, and a required minimum number of ray paths in that orientation. Were there any tests to see if tighter (if possible) or broader bins would change the result significantly? Also, were there any tests to check how the results might vary if different limits for the 30°-bins are being chosen (e.g., 40°-70°, 50°-80° or something like that)?

We thank R2 for this helpful comment. We have indeed experimented with our averaging scheme and now explore its impact in detail in the new Supplementary Figure S2. The figure shows that changing the averaging scheme leaves the main interpreted features intact, both when varying the bin edges (e.g., 0°-30°, 10°-40°, 20°-50°; top row of Figure S2) and when changing the size of the azimuthal swath (15° and 45°; middle row of Figure S2). It speaks to the robustness of the interpretation, how stable the features remain across these different averaging schemes. We thank R2 for motivating us to include these detailed tests. We now refer to Figure S2 at line 44 of the revised manuscript.

I. 39: “2°x2°”

A slightly similar comment to the one above. Have there been any tests with different grid sizes?

Yes, we have experimented with different grid sizes and found this to be an important parameter. In response to R2’s comment, we have added results for two additional grid sizes (1° × 1° and 4° × 4°) in the Supplementary Information (Figure S2, bottom row). For the fine grid, fewer than five measurements fall into many bins (see procedure described in the main manuscript), which limits coverage in some key parts of our study region. The coarse grid, on the other hand, tends to smooth out the deformation streaks seen in the main manuscript, as it averages across them, although the larger-scale features remain clearly visible. Overall, our results appear robust, and we consider the 2° × 2° grid used in the main manuscript to represent the best compromise between resolution and coverage. With the new Supplementary Figure S2, readers can now assess this for themselves.

Figure 1:

- *“blue” – I do not know if this is an issue with the document, but the blue lines in Figures a and b appear purple in my version.*

We thank R2 for noting this -- the lines are indeed violet. They were blue in an earlier version of the figure, and the caption was inadvertently not updated after the revision.

- *“black line majority of analyzed models show low velocities” – this explanation could be made clearer in my view by using the same caption as in Supplementary Figure 2a (“Black lines show where three out of five models indicate low velocities in a cluster analysis”). If not, it would at least be good to have a colon behind “black line”.*

Thanks, we agree and now use a similar wording as in (the previous) Supplementary Figure 2a.

- *“Bottom: Same as left panel” – I think this should have been “Bottom: Same as top panel”*

Yes, thanks for catching. We have corrected the wording.

Figure 2:

- *Is there a reason the surface has been smoothed? Supplementary Figure 1 shows the 2°x2° grid pixels, so a similar thing could be reproduced here (just to avoid the fact that features are misinterpreted due to changes in their shapes as a result of the smoothing).*

R2 raises a good point. We agree that it is important to verify that smoothing does not distort the results. While we find the smoothed version of the figure to be more visually appealing and a faithful representation of the unsmoothed data, this was not evident in the previous version of the manuscript, as the unsmoothed results were not shown. We have now added a new Supplementary Figure S3a displaying the unsmoothed results, allowing readers to assess this directly. The caption of Figure 2 in the main manuscript has also been updated to reference this new figure.

- *Here and also especially in Supplementary Figure 2: are there any uncertainties on the values? It could be helpful to see anything like that to estimate the robustness of the features presented and discussed in this study.*

This is a great suggestion – we now show bootstrap uncertainties in the new Supplementary Figure S3b. This uncertainty is ~ 0.1 on average in our region of interest, which is small compared to the magnitude of the interpreted δS_I values. We now mention these uncertainties in the caption of Figure 2.

Figure 3:

- *Have there been models with different shapes of the feature, other than what appears to be a square one? Like in the comment above, this again could be interesting for the robustness estimation of the features.*

We thank R2 for this comment! We have run a significant number of new models in response to the comments for R1, with much larger model size, different velocity boundary conditions, and different shapes of the thermochemical piles. These new models confirm our original results that convergent flows cause linear streaks of strong deformation. Additionally, the new models show that a unidirectional migration of the Perm Anomaly may not lead to significant changes of the pre-existing deformation field about the Anomaly. As a result, our seismic anisotropy may represent frozen-in deformation of past origin. At the end of the revised manuscript, we now mention more explicitly that our regional models are much simplified compared to the real Earth. Our goal is to use these simple models to explore a specific, hypothesized mechanism that could explain our seismic observations and not to reproduce all complexities of the real Earth. We now discuss these new and revised geodynamic results in detail in lines 94-137 of the revised manuscript.

l. 162: “ Δt ” – this should be “ ΔT ”.

Thanks, corrected.

*David Schlaphorst
Lisbon, 27/10/2025*

Thanks again for your comments and suggestions, which have greatly helped improve the paper.

We have revised our manuscript our manuscript, "Mantle deformation records fossil convergent upwelling at Perm Anomaly", for publication at Nature Communications.

We sincerely thank the reviewers for their time and their positive overall assessment of our work. We are particularly encouraged that Reviewer #2 (R2) found the study robust and is happy with the revised version, and that Reviewer #1 (R1) found our new geodynamic modeling experiments to be an "elegant and appropriate" addition to the study.

Below we have reproduced the comments from you and the reviewers verbatim in *black, italicized* text, with our responses in green color and regular font. Where useful, we denote line numbers in the revised manuscript where changes were made, which are in green font in the new manuscript's revised text.

Reviewer #1 (R1)

Review of 'Mantle deformation records fossil convergent upwelling at Perm Anomaly' submitted to Nature Communications by Jonathan Wolf, Mingming Li, and Barbara Romanowicz

The authors have carefully considered reviewer comments and made changes accordingly, including running new mantle flow simulations that suggest that the pattern of anisotropy associated with the initial upwelling of the Perm Anomaly could be retained after at least 25 million years of lateral motion (Figure 3b). These new experiments are an elegant and appropriate way to address my original comments. Is there a reason why the results are only shown after 25 million years, and not for later times? The size of the new ('Type-2') models is larger than the size of the original ('Type-1') models, making it possible to consider a phase of lateral advection. The half-width of the model (~6,600 km) and the migration velocity (2 cm/yr or 22 km/Myr) determine the duration for which a phase of lateral advection can be considered (maximum $6,600/22 = 300$ Myr). In practice, that duration is shorter because edge effects would be likely to affect the flow and the results after some time (depending on the flow condition on the left-hand side; an open flow boundary condition is, in principle, possible but arguably not necessary in this case). However, 25 million years is much shorter than 300 million years. It would be helpful to present and discuss the results of the models after a longer phase of lateral advection – ideally up to 150 million years. At which stage (if any) would a pattern of anisotropy associated with lateral advection be expected to develop? How long could 'frozen-in' patterns of anisotropy associated with the upwelling of a Perm-like anomaly be retained for? Could this information be used to speculate about the evolution of the Perm Anomaly (its possible history of upwelling and lateral advection)?

We thank R1 for this constructive suggestion. Following this recommendation, we have extended the model runs to substantially longer timescales to track the evolution of the anisotropy. We now present strain patterns at 25, 48, and 98 Myr after the onset of westward motion (Fig. 4b, Supplementary Fig. S9b, and Supplementary Fig. S9c, respectively).

Our results show that although the pile itself is advected westward, the previously developed high-strain pattern persists for several tens of Myr. By ~100 Myr, however, the strain field has evolved significantly, resulting in a markedly different pattern (Supplementary Fig. S9c). This ~100 Myr duration effectively answers the question of how long "frozen-in" patterns can be retained in this specific geodynamic model. The gradual modification of the strain pattern arises because the imposed boundary condition drives unidirectional motion only at the CMB; in the

overlying regions, the flow deviates from pure westward translation due to the mantle's intrinsic dynamics. We have added a discussion of these longer-term results and their implications for the Perm Anomaly's history to the revised manuscript, see lines 112-122.

Crucially though, as now discussed at lines 138-141, the persistence of the strain patterns depends on whether the flow is unidirectional throughout the mantle column. If mantle flow were unidirectional and uniform at all depths, no additional deformation would occur, and the previously developed strain pattern would remain entirely unchanged. Consequently, we propose that the observed seismic anisotropy is likely at least partially of fossil origin, possibly representing a strain field established >100 Myr ago that has been effectively "frozen in" during lateral advection. However, because these patterns can be retained without significant modification during the translation phase, the patterns themselves do not uniquely constrain the timing of the deformation event. But they do serve as evidence that the anomaly has undergone minimal internal deformation since its initial formation.

In the abstract, consider changing 'likely' to 'possible'

We thank the reviewer for this observation. We agree that it is important to avoid overstating the certainty of the model predictions. In line with this feedback, we have revised the abstract to describe the scenario as "plausible" rather than "likely." This terminology accurately reflects that the results are consistent with our geodynamic constraints, while acknowledging the inherent non-uniqueness of the model solutions.

L. 44–45: consider rephrasing to 'The results are robust to alternative averaging schemes and associated uncertainties (Supplementary Figures S2 and S3).'

This is a good suggestion; we have rephrased the sentence accordingly.

L. 126: consider changing 'subtleties' to 'properties'

We agree that 'subtleties' could be improved; however, we feel that 'complexities' serves as a more descriptive alternative than 'properties' in this specific context.

Nicolas Flament, Wollongong, 22 January 2026

Reviewer #2 (R2)

This is the revised version of the manuscript that I have reviewed before, so I will mainly focus on the authors' responds to my comments and the changes in general. I think the authors put effort into answering the suggestions of both reviewers to improve the manuscript. If those are sufficient, I can only confidently answer for my suggestions; for the other reviewer's comments it would be better to have their feedback. It is good to see that the additional tests shown in the added supplementary figures added to the robustness of the study overall and I am happy with the revised version.

We sincerely thank R2 for their time in re-evaluating our manuscript and for their positive assessment of the revised version. We are very pleased to hear that the additional tests and supplementary figures successfully addressed your concerns and contributed to the overall robustness of the study. We appreciate your acknowledgment of the effort put into the revision; your constructive feedback throughout this process has significantly improved the quality of our work.

We have revised our manuscript, "Mantle deformation records fossil convergent upwelling at Perm Anomaly", for publication at Nature Communications. We sincerely thank R1 the positive overall assessment of our work.

Below we have reproduced the comments from you and the reviewers verbatim in *black, italicized* text, with our responses to the remaining comment in green color and regular font. Where useful, we denote line numbers in the revised manuscript where changes were made, which are in green font in the new manuscript's revised text.

Reviewer #1 (R1)

Review of 'Mantle deformation records fossil convergent upwelling at Perm Anomaly' submitted to Nature Communications by Jonathan Wolf, Mingming Li, and Barbara Romanowicz

The authors have elegantly addressed my remaining comment by showing results after 48 Myr and 98 Myr of imposed westward flow at the CMB (Fig. S9b–c). I look forward to seeing this work published. If they wish to do so, the authors could moderate the text on L. 112–122 at the proof stage. Two end-member mantle flow scenarios are mentioned: 'unidirectional motion only at the CMB' (L. 120–121; scenario implemented in the model) or 'mantle flow [] unidirectional and uniform at all depths' (L. 117–118). Some text clarifying that neither scenario is likely to be representative of mantle flow from ~100 Ma around the Perm Anomaly would be helpful. Notwithstanding the complex interaction of sinking slabs with mantle upwelling, 'mantle flow [] unidirectional and uniform at all depths' seems incompatible with the viscosity contrast between the upper mantle and lower mantle inferred from studies of the geoid and post-glacial rebound: lower mantle flow is likely to be slower than upper mantle flow.

Nicolas Flament, Wollongong, 20 February 2026

We thank the reviewer very much for his comment. We have implemented his suggestions, and now mention that our modelling addresses two end-member scenarios with the real Earth likely being more complex for the reasons that the reviewer has mentioned above (lines 112-121).